# T Cells Expressing a TCR-Like Antibody Selected Against the Heteroclitic Variant of a Shared MAGE-A Epitope Do Not Recognise the Cognate Epitope

**DOI:** 10.3390/cancers12051255

**Published:** 2020-05-16

**Authors:** Mesha Saeed, Erik Schooten, Mandy van Brakel, David K. Cole, Timo L. M. ten Hagen, Reno Debets

**Affiliations:** 1Laboratory of Experimental Oncology, Department of Pathology, Erasmus MC, 3000 CA Rotterdam, The Netherlands; m.saeed@erasmusmc.nl; 2Laboratory of Tumor Immunology, Department of Medical Oncology, Erasmus MC Cancer Institute, 3000 CA Rotterdam, The Netherlands; schooten@linxispharmaceuticals.com (E.S.); m.vanbrakel@erasmusmc.nl (M.v.B.); j.debets@erasmusmc.nl (R.D.); 3Division of Infection and Immunity and Systems Immunity Research Institute, Cardiff University School of Medicine, Heath Park, Cardiff CF14 4XN, UK; coledk@cardiff.ac.uk

**Keywords:** peptide:major histocompatibility complex, phage display screening, chimeric antigen receptor, T cell engineering, heteroclitic peptide

## Abstract

Antibodies-recognising peptides bound to the major histocompatibility complex (pMHC) represent potentially valuable and promising targets for chimeric antigen receptor (CAR) T cells to treat patients with cancer. Here, a human phage-Fab library has been selected using HLA-A2 complexed with a heteroclitic peptide variant from an epitope shared among multiple melanoma-associated antigens (MAGEs). DNA restriction analyses and phage ELISAs confirmed selection of unique antibody clones that specifically bind to HLA-A2 complexes or HLA-A2-positive target cells loaded with native or heteroclitic peptide. Antibodies selected against heteroclitic peptide, in contrast to native peptide, demonstrated significantly lower to even negligible binding towards native peptide or tumour cells that naturally expressed peptides. The binding to native peptide was not rescued by phage panning with antigen-positive tumour cells. Importantly, when antibodies directed against heteroclitic peptides were engineered into CARs and expressed by T cells, binding to native peptides and tumour cells was minimal to absent. In short, TCR-like antibodies, when isolated from a human Fab phage library using heteroclitic peptide, fail to recognise its native peptide. We therefore argue that peptide modifications to improve antibody selections should be performed with caution as resulting antibodies, either used directly or as CARs, may lose activity towards endogenously presented tumour epitopes.

## 1. Introduction

T cells are key players in the adaptive immune response, also in cancer, and are equipped with T-cell receptors (TCRs) that recognise short peptide complexes presented by MHC molecules. CD8 cytotoxic T lymphocytes (CTLs) play a critical role in the eradication of tumour cells and recognise peptides in the context of MHC class I molecules (MHCI). Adoptive T-cell therapy (AT), a well-tested and promising approach to treat cancer, relies on the infusion of autologous tumour-specific T cells. Besides the use of non-modified T cells, such as tumour-infiltrating lymphocytes (TILs) or peripheral T-cell clones, there is a clear shift toward the use of T cells that are gene-engineered to express chimeric antigen receptors (CARs). These CAR-engineered T cells recognise a chosen tumour antigen, and are redirected to selectively destroy cells expressing this antigen. The use of CARs has shown impressive results in B-cell leukaemia with response rates up to 94% [1], which culminated in FDA approvals of *Kymriah* [2] and *Yescarta* [3]. The success of CAR T cells in the treatment of solid tumours, however, lags behind its success in haematological tumours. TCRs have a natural ability to recognise the full spectrum of tumour cell-derived peptides, whereas CARs only recognise extracellular antigens, and may provide T cells with a therapeutic advantage when treating solid tumours. Indeed, TCR-engineered T cells have demonstrated clinical benefit in patients with multiple myeloma, but also metastatic melanoma and metastatic synovial sarcoma with response rates varying between 55 and 80% [4].

Antibodies specific for peptide: MHC complexes (pMHC) with the intent to target tumour cells either directly or via CAR-engineered T cells would potentially widen the therapeutic window to treat solid tumours. It has been shown previously that, using phage antibody display technology, it is possible to obtain antibodies that recognise tumour-derived peptides in an MHC restricted manner [5,6,7,8,9]. These so-called ‘TCR-like’ antibodies harbour the same specificity as a TCRs, while binding affinities are generally higher. Such antibodies may serve as improved reagents for imaging or immune therapies, including CAR T-cell therapies. In order to increase the efficacy of antibody selections, heteroclitic peptides have been used to improve binding to and stability of MHC. Heteroclitic peptides are generally peptides that contain favourable residues at positions that anchor into the peptide-binding groove of MHC. Over the past years, vaccinations using heteroclitic peptides from tumour-associated antigens, such as RAS [10], melanoma antigen recognised by T cells 1 (MART1) and glycoprotein 100 (gp100), have been widely employed with the intent to treat cancer patients [11,12,13,14]. Following treatment, patients often did show enhanced frequencies of anti-vaccine T cells, yet these changed T cell frequencies were mostly not accompanied by clinical responses. Despite the fact that the use of heteroclitic peptides is a recognized strategy for vaccinations, recent studies have shown that anchor residue-modified peptides can be less effective than their cognate counterparts [15,16], or do not result in enhanced immune responses [17]. With respect to underlying mechanism of action, there is still discordance among various studies, as some argue that anchoring amino acids to the MHC molecule do not affect recognition by T cells as only centrally located, extending amino acids in the peptide, are recognised, whereas others argue that buried or anchoring peptides do affect recognition by the TCR [18].

We questioned whether MHC-restricted antibodies selected against heteroclitic or native peptides improve the therapeutic potential of gene-engineered T cells without loss of tumour specificity. It is now well established that tumour cells express antigens that are presented via MHCI molecules, and are recognised by CTLs derived from cancer patients [19,20,21], such as gp100, melanoma associated antigen (MAGE), New York oesophageal squamous cell carcinoma (NY-ESO), carcinoembryonic antigen (CEA) and cell tumour antigen (p53) [22,23,24,25]. Antibodies that are selected against peptide:MHC complexes have been reported to display TCR-like functionality, limited off-target binding and increased target specificity [26,27]. Here, we study novel antibodies that recognise HLA-A2-restricted peptides that are shared among multiple MAGEs. MAGEs may provide antigen targets with therapeutic potential as the expression of some MAGEs are silenced in healthy adult tissues, while in testis, these antigens are present in the absence of MHC [28]. A multi-MAGE (mMA) epitope was first described by Graff-Dubois et al. [12] and constitutes a heteroclitic peptide (p248V9) that is shared by multiple MAGE-A genes (see Figure 1A) and would provide a target for multiple tumour types. In the current study, we have used a large human phage-Fab library to select TCR-like antibodies against the heteroclitic multi-MAGE peptide p248V9 as referred in [12], and demonstrate that these antibodies show binding towards the modified pMHC complex but are not able to recognise native peptides nor antigen-positive tumour cells neither directly nor as a CAR that is expressed on the surface of T cells. These findings do not warrant general use of heteroclitic peptides to establish immune therapies.

## 2. Materials and Methods

### 2.1. Cell Lines

The EBV-transformed B-cell line BSM and LCL/T-lymphoblastoid hybrid cell line T2 were maintained in RPMI supplemented with 10% FBS, 1% penicillin/streptomycin and 1% L-glutamine. The melanoma tumour cell lines DAJU, MZ2Mel43, the oesophageal carcinoma cell line TE-4 and the bladder carcinoma cell line OBR were maintained in Dulbeco’s modified Eagle’s medium (DMEM) supplemented with 1% penicillin/streptomycin, 1% L-glutamin and 1% non-essential amino acids. DAJU and MZ2Mel43 were kindly provided by Pierre van der Bruggen (Ludwig Institute of Cancer Research, Brussels, Belgium) and OBR was a kind gift from Graff Dubois (Institut National de la Sante et de la Recherche Medicale, Institut Gustave Roussy, Paris, France). Tumour cells were checked for gene-expression of MAGE-A3, A6, A10, A12 and HLA-A1 and A2 (primer sequences are provided upon request), as well as surface expression of MAGE-A3(FLW) peptide in the context of HLA-A2.

### 2.2. Peptides and pMHC Complexes

To evaluate the quality of the human phage-Fab library (see below), the following proteins have been used for selections: negative regulatory factor (NEF**_56–205_** accession number CAA13474.1), latent membrane protein 1 (LMP1 accession number QAR15098.1) and prostate specific membrane antigen (PSMA accession number AAA60209.1) (sequences listed in Appendix A). All pMHC complexes (biotinylated HLA-A2 monomers,) and peptides were obtained from Sanquin (Amsterdam, The Netherlands). Peptides used are: multi-MAGE wild-type peptide (mMA-wt, YLEYRQVPG); mMA heteroclitic peptide (mMA-GV, ELEYRQVPV); and Influenza peptide (INF, CTELKLSDY). HLA-A2-complexed peptides used are: heteroclitic multi-MAGE peptide (mMA-GV/A2); prostate specific antigen-derived peptide (PSA/A2, KLQCVDLHV); and prostate specific membrane antigen-derived peptide (PSMA/A2, LLHETDSAV).

### 2.3. T2 Assay

T2 cells were used to assess peptides for their binding by cellular HLA-A2 as described previously [16,18]. In short, T2 cells were pulsed with peptides at concentrations 1 and 50 mM, incubated at 26 °C for 14–16 h and then at 37 °C for 2 h. HLA A*0201 surface expression was determined through staining with a PE-labelled antibody (BB7.2, BD Biosciences, San Jose, CA, USA). All samples were processed in duplicate and analysed for mean fluorescence intensity (MFI) of HLA-A2 using FlowJo Software (TreeStar, Ashland, OR, USA) on BD FACSCanto™ II flow cytometer (Becton, Dickinson Company, Franklin Lakes, NJ, USA). MFI values of mMA and other peptides were normalised against those observed with Influenza peptide.

### 2.4. Construction of Phage-Fab Display Library

Unless mentioned otherwise, Fab libraries were built according to the procedure previously described by de Haard et al. [29]. Two different antibody sources were used, namely peripheral blood lymphocytes (PBLs) derived from buffy-coats from four healthy donors and isolated using Ficoll–Paque gradient centrifugation as well as a human spleen cDNA library already containing 5 × 10^9^ clones (Superscript human spleen cDNA library, Invitrogen, Carlsbad, CA, USA). Total RNA from PBLs was isolated using TRIzol (Invitrogen), and variable region genes were reverse-transcribed and amplified by PCR, followed by digestion of VH as well as VκCκ and VλCλ amplicons and their cloning into the pCES1 phagemid vector. Subsequently, the spleen-derived Vκ and Vλ libraries were sub-cloned into the pCES1 vector that already contained the spleen-derived VH library, and the PBL-derived VH library was sub-cloned into the pCES1 vector that already contained the PBL-derived Vκ or Vλ libraries. In parallel, the VH library derived from spleen was combined with the Vκ and Vλ libraries from PBL, and the VH library derived from PBL was combined with those of Vκ and Vλ from spleen. In all sub-clonings, vector fragments were gel purified using Zymoclean (Baseclear, Leiden, the Netherlands). Ligations were performed O/N at 16 °C, after which mixtures were ethanol precipitated, and introduced into Escherichia coli TG1 cells by electroporation. Phagemid particles were rescued on a large scale using M13K07 helper phages as explained elsewhere [5,30]. Phages were precipitated and 0.45 µm filter sterilised and stored at −80 °C in PBS containing 15% glycerol. Prior to each selection, phages from Fab libraries and combinatorial Fab libraries were pooled in relation to their library size to ensure that the diversity of each library was covered and each clone was represented at least 100 times (see Appendix A).

### 2.5. Selection of TCR-Like Antibody-Expressing Phages

Phages (10^13^ colony forming units; cfu) were pre-incubated for 1 h at RT in PBS containing 2% non-fat dry milk (PBSM). In parallel, 200 μL streptavidin-coated beads (Dynal) were equilibrated for 1 h in PBSM. For subsequent panning rounds, 100 μL beads were used. To deplete the phage-Fab library for pan-MHC binders, 200 nM of biotinylated pMHC complexes containing an irrelevant peptide were added prior to each selection round and incubated for 30 min under rotation, after which, 100 μL of the equilibrated beads were added, and the mixture was incubated for another 15 min under rotation. Beads were magnetically removed, and the remaining phage-Fab fraction was used for first round selections with 200 nM of biotinylated pMHC and incubations of 1 h at RT under continuous rotation. For the second and third selection rounds, 20 nM of pMHC was used. A depletion step against an HLA-A2 complex bearing an irrelevant peptide (‘irrelevant pMHC’) was included in each panning round with the intent to remove any HLA-A2-binding Fabs.

To limit the isolation of phages only binding to HLA-A2, a soluble HLA-A2 binding Fab, named D3, previously isolated from the same library, was added to the phage-pMHC mixture during incubation (50, 10 and 5 µg for rounds 1–3, respectively). Non-bound phages were removed by 5 washing steps with PBSM, PBS and 0.1% Tween (PBST) and again PBS. Phages were eluted from the beads by adding 500 μL freshly prepared 100 mM triethylamine (TEA) and incubating for 10 min under rotation. Neutralisation was performed by addition of 500 μL 1 M Tris pH 7.5. Following selection, eluted phages were incubated with exponentially growing E. Coli TG1 cells for 30 min at 37 °C and infected TG1 cells were plated on YT-agar plates. Next day, colonies were harvested, and an inoculum, with volume representing the size of the selected library, was used for a subsequent selection. For phage library selections using biotinylated proteins, the same procedure was followed except for the depletion step with irrelevant pMHC and addition of soluble Fab D3. For selections using MAGE-A positive, HLA-A2 positive DAJU cells, 1 × 10^6^ cells were harvested and pre-blocked for 1 h at RT in PBSM containing 10% FBS (PBSM/FBS) while rotating. Simultaneously, phages were also pre-incubated in PBSM/FBS. After 1 h, cells were spun down and re-suspended with pre-incubated phages and mixed for 1 h at RT while rotating. The cell-phage mixture was then washed 10 times with PBSM/FBS, followed by two washing steps with PBS. Phages were eluted with TEA for 10 min and the mixture was neutralised with Tris. After neutralisation, lysed cells were spun down and the supernatant was used to re-infect E. Coli TG1 cells.

### 2.6. Testing Phages for pMHC Binding, Clonality and Binding to Target Cells

Randomly selected phage clones were assessed for pMHC binding via ELISA. To this end, phages were produced in 96-wells plates using M13 k07ΔpIII (hyperphage; Progen, Heidelberg, Germany). Streptavidin pre-coated 96-well ELISA plates (SanBio, Uden, The Netherlands) were coated overnight (4 °C) or 1 h at RT with 0.1 μg/mL biotinylated pMHC in PBS. MHC complexes with irrelevant peptide served as negative controls. Plates were blocked with PBSM for 30 min at RT and incubated with supernatant of single phage clones in PBSM for 1 h at RT. After extensive washing with PBST, plates were incubated with horseradish peroxidase-conjugated anti-M13 antibody (Thermo Scientific, Waltham, Massachusetts, United States), and detection was performed using TMB reagent (Sigma, Zwijndrecht, The Netherlands) [5]. DNA fingerprinting was used to assess the diversity of antibody clones that specifically bound to mMA-GV/A2 (OD ratio between mMA-GV and irrelevant pMHC > 30). Briefly, DNA of individual phage clones was digested with BstN1 for 1 h at 37 °C, run on 2% agarose gel at 100 V for 60 min and visualised under UV light. Lastly, phages containing unique clones were tested for binding towards pMHC-expressing target cells. In short, 0.5 × 10^6^ target cells were incubated with 1 × 10^10^ cfu of phages. T2 and BSM cells were loaded with 10 μM peptide for 30 min at 37 °C prior to staining. Next, cells were washed and incubated with 100 μL of 1 µg/mL mouse anti-M13 antibody and detected with PE-conjugated goat anti-mouse antibody using FACSCalibur (Becton, Dickinson, Franklin Lakes, NJ, USA), and analysis was done by FCS express 4.0 (De Novo Software, Los Angeles, CA, USA).

### 2.7. Generation of CAR T Cells

VH and VL chains of the AH5 antibody clone, specific for mMA-GV/A2, were genetically fused via a flexible linker and were provided SfiI and NotI restriction sites (products were gene-synthesised at Geneart, Regensburg, Germany). The resulting single-chain Fv (scFv) was cloned into the retroviral vector pBullet [31], thereby fusing the scFv to the CD28 transmembrane domain and the intracellular Fc(γ)RIγ signalling domain as described previously [31]. The resulting AH5 CAR:28γ was sequence verified, retrovirally introduced into human T lymphocytes from two healthy donors as described before [32] and sorted with FACSAria II (Becton, Dickinson Company, Franklin Lakes, NJ, USA) using mMA-GV/A2 tetramers, resulting in >90% of T cells expressing mMA-GV/A2 CAR (as determined by flow cytometry).

### 2.8. Testing CAR T Cells for Specificity and Responsiveness

CAR T cells (6 × 10^4^) were cultured for 24 h either in the presence or absence of target cells (2 × 10^4^) that were or were not loaded with peptides as described [31]. Culture medium was supplemented with 360 IU/mL rIL-2. Supernatants were harvested and levels of IFNγ were measured by ELISA according to the manufacturer’s instructions (Sanquin, Amsterdam, The Netherlands).

### 2.9. Statistical Analyses

All experiments were performed at least in duplicate or triplicate, and data were processed using GraphPad Prism. Statistical analysis was done using Kruskal–Wallis and Mann–Whitney *u* tests. Outcomes with *p* - values less than 0.05 were considered significantly different.

## 3. Results

### 3.1. Combinatorial Phage-Fab Library with High Level of Diversity Enables Selection of TCR-Like Antibody Clones

Using sequential cloning, we have combined PBL and spleen-derived heavy and light chain libraries and created a non-immunised Fab library with 5.6 × 10^9^ individual clones of which 78% have a full-length Fab insert. An overview of the size of both single chain as well as Fab libraries, and percentage of inserts is given in Appendix A. In addition, combinatorial libraries were generated by exchanging VH and VL chains from PBL and spleen-derived repertoires. The sizes of these combinatorial Fab libraries are about 3 × 10^9^ individual clones of which 87% have a full-length Fab insert. Finally, when PBL, spleen and combinatorial libraries were all combined, a single Fab library was created with a size of 8.6 × 10^9^ individual clones (corrected for percentage of full-length Fab inserts). Selections with various control proteins (NEF, LMP1e and PSMA) were performed to test the applicability of our newly generated phage-Fab library (Appendix A). Already after two to three rounds of selection against these proteins we obtained multiple ELISA-positive binders and unique clones (the latter according to DNA restriction patterns). We then used wild type and heteroclitic mMA peptides for further experiments (see also Figure 1A). As shown in Figure 1B, binding of mMA-wt to HLA-A2 is equal to the setting with no external peptide, whereas mMA-GV shows clear and titratable binding to HLA-A2. Because of none to negligible binding of mMA-wt towards HLA-A2, (Figure 1B) this complex could not be produced as a stable soluble pMHC, yet we did include melanoma cells natively expressing mMA-wt in our library selections (see below and Table 1).

### 3.2. Phage-Fab Selections Against Heteroclitic mMA Peptide Revealed Unique AH5 Clone

We used the phage-Fab library to isolate TCR-like antibodies against mMA-GV/A2, and after three selections rounds, the output titer showed an enrichment of more than 2000 times in comparison with the first selection round (Table 1). Of the 282 analysed phage-Fab clones, 46 (16%) specifically bound to mMA-GV/A2 and not to an irrelevant peptide/A2 complex. After DNA fingerprinting, 18 out of these 46 binders showed unique DNA restriction patterns (Table 1). To identify the best binders, a titration ELISA using recombinant mMA-GV/A2 complexes was performed with these 18 unique clones (Figure 2A). Six binders that still showed clear binding at relatively low phage titers (<1 × 10^6^) were selected for further evaluation using HLA-A2-positive BSM cells pulsed with mMA-GV or irrelevant peptide (Figure 2B). Only two clones, AH5 and CB1 showed clear and reproducible binding to BSM cells when pulsed with mMA-GV peptide.

In a next step, we identified HLA-A2-positive tumour cells from various histological origins that express MAGE-A3, A6, A10 and/or A12 to establish a panel of tumour cell lines with endogenous expression of mMA (Figure 3A). All tumour cells in this panel express at least one, but mostly multiple MAGE-A genes. Interestingly, DAJU cells express all analysed MAGE-A genes. To verify whether these tumour cell lines did not have a defective antigen processing and presentation machinery, we stained the cells using an antibody directed against a MAGE-A3 (FLW)-derived peptide in the context of HLA-A2. All tumour cell lines, except for the HLA-A2-negative MZ2Mel43, demonstrated positive staining, indicating that MAGE-A-derived peptides are presented via HLA-A2 (Figure 3B). We then adapted the above selection procedure by including MAGE-A and HLA-A2-positive DAJU cells at either the second or fourth selection round (Table 1). Panning on DAJU cells in the second round followed by two additional panning rounds on mMA-GV/A2 led to a strong enrichment in phage output titer (>2000) and five unique mMA-GV/A2-specific clones were identified. Interestingly, one clone was identical to AH5. Panning on DAJU cells in the final fourth round not followed by additional panning rounds on mMA-GV/A2 showed a strong drop in phage output titer and yielded 14 mMA-GV/A2 specific binders, from which 9 unique clones were identified. Also, here one of these clones was identical to AH5. Unexpectedly, these additional clones (besides AH5) obtained after selection strategies in which cells were included, did not bind to peptide-pulsed BSM cells.

### 3.3. mMA-GV/A2-Specific AH5 Antibody Does Not Recognize mMA wt Peptide nor mMA-Positive Tumor Cells

Characterisation of phage-AH5 using various pMHC complexes showed that AH5 only binds to mMA-GV/A2 and not to any of the other pMHC complexes (Figure 4A). Next, TAP deficient T2 cells were pulsed with serially diluted mMA-wt or mMA-GV peptide, postulating that when using a TAP deficient cell line, no competition with endogenously processed peptides will occur, thereby enhancing the sensitivity of the assay. Again, no binding of AH5 was observed towards mMA-wt-loaded T2 cells. In contrast, AH5 strongly bound to T2 cells loaded with mMA-GV peptide in a concentration-dependent manner (EC50 = 1.25 ug/mL or 0.95 mM) (Figure 4B). To provide further proof that AH5 is not able to recognise endogenously processed mMA peptide displayed by HLA-A2, we extended the flow cytometric experiments using the panel of tumour cells described above and BSM cells. No detectable shift in fluorescence intensity was observed with any of the tumour cells (Figure 4C); a finding that is irrespective of IFNγ pre-treatment to increase HLA-A2 surface expression levels. In contrast, BSM cells pulsed with mMA-GV but not wt peptide showed a complete shift when stained with phage AH5.

### 3.4. AH5 CAR Expressing T Cells Do Not Respond to Tumor Cells Endogenously Presenting mMA

Finally, we assessed whether a heteroclitic peptide-selected antibody would be able to mediate a response towards the cognate epitope in a more sensitive and therapeutically applicable system. To this end, scFv AH5 was expressed as a chimeric antibody-based receptor (CAR) on the surface of T cells. Upon retroviral transduction of AH5 CAR into primary human T lymphocytes, surface expression was determined by staining with mMA-GV/A2 tetramers (Figure 5A). Almost all CAR T cells (94%) expressed AH5 CAR and these T cells did not bind irrelevant gp100-AV/A2 tetramers (Figure 5B). T2 loaded with mMA-GV peptide were able to induce a CAR T cell IFNγ response in a concentration-dependent manner (EC50 = 2.8 ug/mL or 2.2 mM) (Figure 5C). In contrast, no IFNγ response was observed when T2 cells were loaded with mMA-wt peptide, even when very high amounts of peptide were used (Figure 5C). Additionally, no CAR-mediated IFNγ production was observed after stimulation with the panel of tumour cell lines expressing mMA (Figure 5D).

## 4. Discussion

In this study, we obtained a TCR-like Fab antibody AH5 against HLA-A2 complexed with a heteroclitic peptide derived from a shared epitope among various MAGE-A antigens, and thoroughly assessed this antibody for its ability to recognise the cognate (non-mutated) epitope. To this end, we have constructed a combinatorial phage-Fab library (diversity: 8.6 × 10^9^ cfu), employed various selection procedures, characterised antibodies by DNA fingerprinting, ELISA and flow cytometry and evaluated antibodies either directly or as CARs expressed by T cells using peptide-loaded target cells as well as tumour cells endogenously expressing the cognate epitope. AH5 showed clear binding to surface-expressed mMA-GV but not wt peptide.

The lack of observed binding towards native target cells might have a two-fold explanation with regard to quantity and quality of the epitope. Regarding the quantity of the epitope, we have shown that mMA-wt peptide has a non to negligible affinity for HLA-A2, therefore it may be very poorly expressed by tumour cells. Low HLA-A2-presentation of mMA-wt by tumour cells, combined with low binding affinity of AH5 to mMA-wt (as suggested by its identification following phage selections on DAJU cells), could have contributed to the negligible detection of AH5 binding to mMA-wt peptide. Graff-Dubois and colleagues, however, showed in their study that CTLs raised against the same heteroclitic mMA-GV peptide are able to respond to endogenously presented peptide [12], indicating that mMA-wt peptide is expressed by tumour cells. One of the tumour cell lines used in their study was OBR, also taken along in our assays to characterise AH5, but also when testing this cell line, no binding of AH5 was observed. Enhancing HLA-A2 expression levels by pre-treatment with IFNγ did not result in increased binding nor CAR T-cell responses. In an effort to explain why in contrast to our study, the study by Graff-Dubois and colleagues shows that the mMA-GV-elicited T cell clones recognise the cognate epitope, we speculate along two major lines. First, in the current study have generated an mMA-GV-specific CAR:28γ that recognises and mediates T-cell function via an antibody-based receptor coupled to CD28 and Fc(ε)RIγ, which is inherently different than epitope recognition and intracellular signalling mediated via a natural T-cell receptor. In fact, the AH5 CAR may display lower affinity for the native peptide and/or mediate suboptimal activation of T-cell signalling components compared to the TCR of the CTL248 used in the above study. Indeed, our observation that AH5 CAR T cells do not respond to mMA-wt peptide corresponds with findings from Buhrman and Slansky [33] arguing that T cells produced against heteroclitic peptide may have lower affinity for the native peptide thereby preventing efficient recognition and elimination of tumour cells. Second, our CAR T cells may display a less differentiated T cell effector phenotype compared to CD8 T cell clones, providing a potential other reason for compromised in vitro functions of the former T cells towards tumour target cells expressing the native mMA epitope.

Regarding the quality of the epitope, only a single amino acid is changed from the mMA-GV peptide sequence at the anchoring site, yet this subtle difference may give rise to different antibodies that cannot, or hardly recognise native peptides [16,34,35]. The assumption that anchor residues are hidden from a T cell’s view is not necessarily in line with our findings, the latter in fact illustrate that the p9 anchor residue is of utmost importance for peptide recognition. Alternatively, peptide modification may change the exposed body of a peptide resulting in selection of a completely different antibody. Cole and colleagues discuss that heteroclitic peptides increase the peptide’s immunogenicity and that buried or anchoring peptides are recognised by TCRs [16]. The use of anchor-modified peptides may alter the binding to MHC and influence T-cell recognition [34] and consequently raise T cells harbouring different TCRs compared to those primed with natural peptides [5]. Our findings are in concordance with these reports and extend these implications to TCR-like antibodies either used directly or in the context of gene-engineered T cells.

With respect to the use of Fabs for the generation of CARs, we would like to put forward the importance of CAR building blocks regarding their immunogenicity as well as expression and function. Because of the chimeric nature of CARs, one cannot exclude the elicitation of an anti-CAR immune response. Already in an early clinical study with autologous T cells expressing a CAR directed toward carbonic anhydrase IX, an antigen highly expressed in renal cell carcinoma, we have reported distinct humoral and cellular anti-CAR responses in combination with limited peripheral persistence of transferred CAR T cells [36,37]. Humoral immune responses were anti-idiotypic in nature and neutralised CAR-mediated T cell function, whereas cellular anti-CAR responses were directed against murine antibody variable domains. Of interest, such immune responses were not detected against boundary regions between antibody fragments and co-stimulatory and/or signalling domains from CD28 and/or Fc(ε)RIγ. Despite the potential occurrence and consequences of such immune responses, reported in multiple studies [38,39], it is important to note that humanised libraries as used in our current study are increasingly being explored to lower the immunogenicity of CARs. In addition, CARs do show differences regarding their in vivo function because of the CAR’s exact format, i.e., inclusion of linkers to space the antigen-recognition unit from the T cell’s plasma membrane, transmembrane domains and/or (multiple) intracellular co-stimulatory domains, such as CD28 (as is the case in the Yescarta product) or 4-1BB (as is the case in the Kymriah product). The varying formats of CARs as well as other aspects of cellular engineering, and their consequences for in vitro and in vivo T cell behaviour are reviewed elsewhere [40,41]; the take home message being that engineering of co-stimulatory CARs into the original TCR locus may provide additional safety and efficacy benefits.

## 5. Conclusions

Our study demonstrates that heteroclitic peptides may provide a non-optimal source for selections of TCR-like antibodies, and warrants caution as such antibodies may show loss of recognition and/or loss of specificity towards the cognate peptide. We argue that heteroclitic peptides, especially when used for clinical purposes, require careful pre-clinical testing to ensure that the elicited cellular and humoral immune response is not exclusively biased towards these anchor-modified peptides.

## Figures and Tables

**Figure 1 cancers-12-01255-f001:**
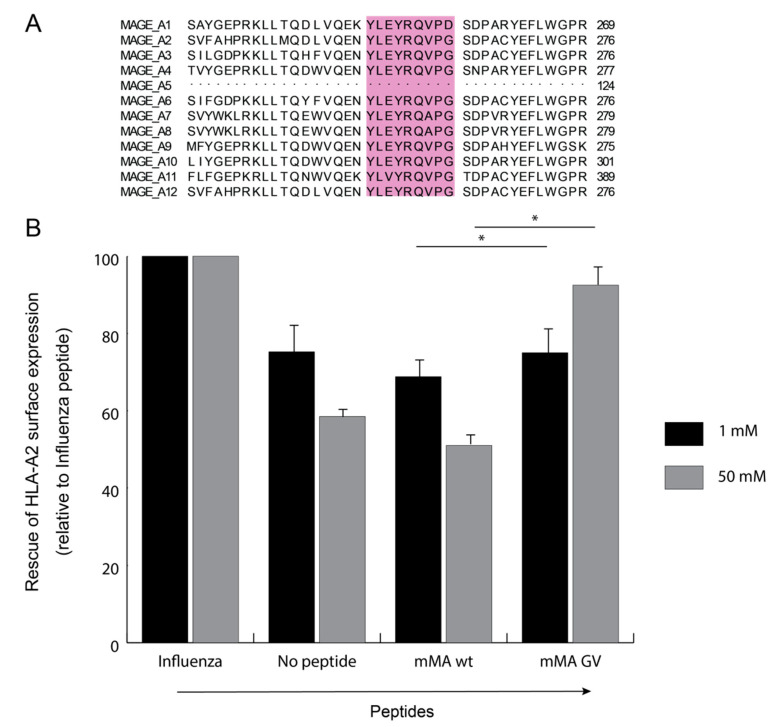
Presence of mMA among various MAGE-A genes and binding of wild type or heteroclitic mMA peptide to HLA-A2. (**A**) Shared 9 amino acid sequence among MAGE A1 to A12 antigens. (**B**) T2 cells were pulsed with titrated amounts of control Influenza (HLA-A*201 restricted influenza M_58-66_ GILGFVFTL peptide), mMA (wt, YLEYRQVPG), mMA (GV, YLEYRQVPV) or no external peptide. Flow cytometry was performed as described in the MM section. MFI values were normalized against those observed with Influenza peptide. Data are presented as mean percentage (at 1 and 50 mM concentrations) and standard error of mean, *n* = 5. Statistical significance is calculated with Kruskal–Wallis test among all groups: 1 mM *p* = 0.007 and, 50 mM *p* = 0.008. Individual non-parametric significance between mMA-GV and mMA-wt is calculated using Mann–Whitney *U* test; *: *p* < 0.05.

**Figure 2 cancers-12-01255-f002:**
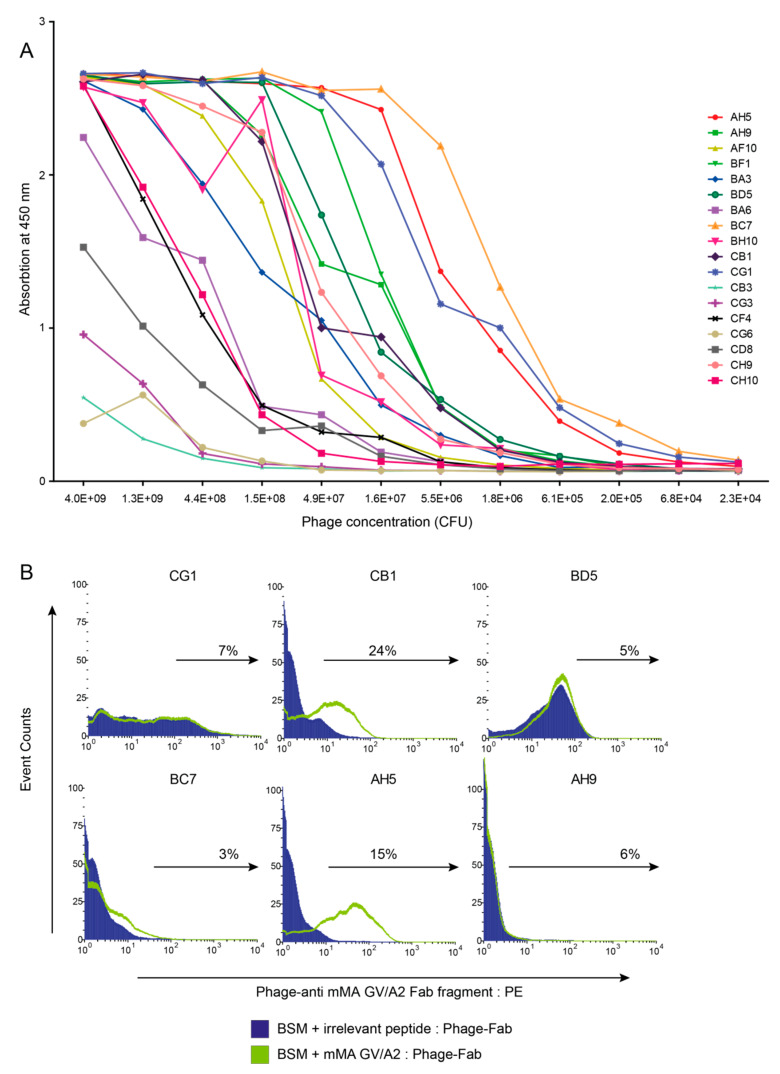
Characterisation of phage-antibody clones for binding to mMA peptides. (**A**) Streptavidin plates were coated with biotinylated soluble mMA-GV/A2. Eighteen phage-antibodies were selected and incubated with titrated amounts of pMHC complexes, and binding was detected per ELISA. A representative example is shown out of two independent experiments. (**B**) Six binders from A were tested for binding towards BSM cells pulsed with mMA-GV or irrelevant peptide by flow cytometry. Data are presented as histograms and percentages in plots indicate binding towards BSM cells loaded with mMA-GV peptide corrected for irrelevant peptides.

**Figure 3 cancers-12-01255-f003:**
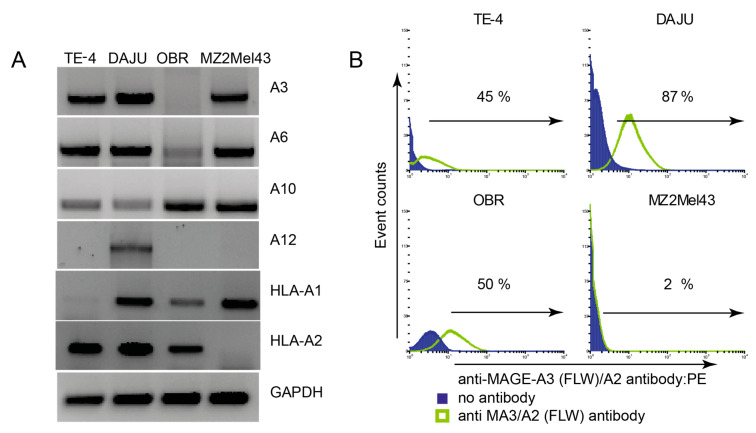
Expression of mMA and HLA by tumour cell lines of different histological origins. (**A**) Expression of various MAGE genes and HLA- A1 and A2 genes are shown together with GAPDH used as a positive control; PCR products were put on agarose gel. (**B**): Tumour cell lines from A were tested for expression of MAGE-A3(FLW)/A2 by flow cytometry. Data are presented as histograms and percentages in plots indicate binding towards tumour cells. Appendix A: Complete PCR Gel showing the expression of MAGE A3 on tumor cell lines, Appendix A: Complete PCR Gel showing the expression of MAGE A6 and MAGE A10 on tumor cell lines, Appendix A: Complete PCR Gel showing the expression of MAGE A6, MAGE A10 and MAGE A12 on tumor cell lines, Appendix A: Complete PCR Gel showing the expression of MAGE A1, HLA A1, HLA A2 and GAPDH on tumor cell lines.

**Figure 4 cancers-12-01255-f004:**
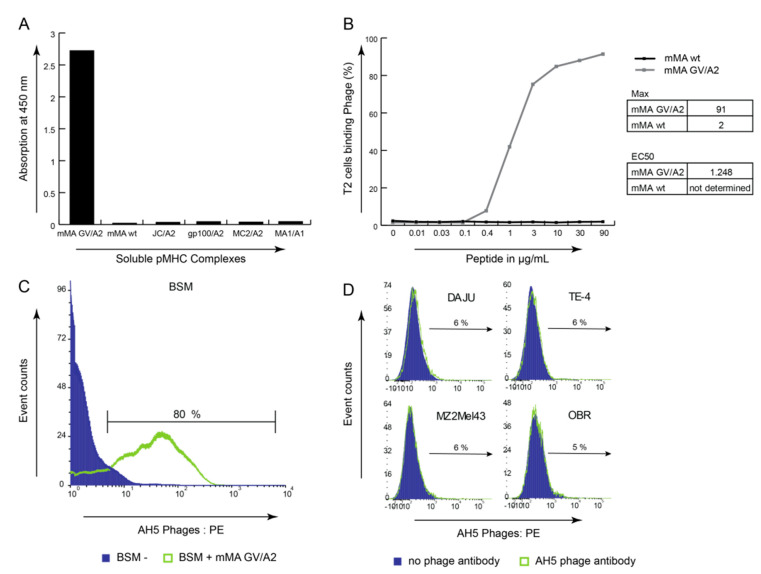
Anti mMA-GV/A2 phage antibody AH5 recognises heteroclitic peptide but not native mMA-wt/A2. (**A**) Phage Fab-AH5 when tested in ELISA specifically recognises mMA-GV/A2 pMHC but not irrelevant pMHC complexes. Data are presented as OD values. (**B**) Phage Fab-AH5 when tested in flow cytometry does bind to mMA-GV but not wt-pulsed T2 cells. T2 cells without external peptide served as a negative control, and data are presented as % T2 cells expressing HLA-A2. EC50 value for mMA-GV/A2 is 1.25 ug/mL or 0.95 mM; calculated by GraphPad Prism and both maximum and EC50 values are displayed. (**C**) Phage Fab-AH5 binds to mMA-GV peptide-loaded BSM cells. Data are represented as histogram overlay, purple and green lines represent HLA-A2-positive BSM cells without external peptide and mMA-GV peptide, respectively. (**D**) Phage Fab-AH5 does not bind to tumour cells expressing mMA-wt peptide. Data are represented as histogram overlays, purple and green lines represent tumour cells stained without or with phage-AH5, respectively.

**Figure 5 cancers-12-01255-f005:**
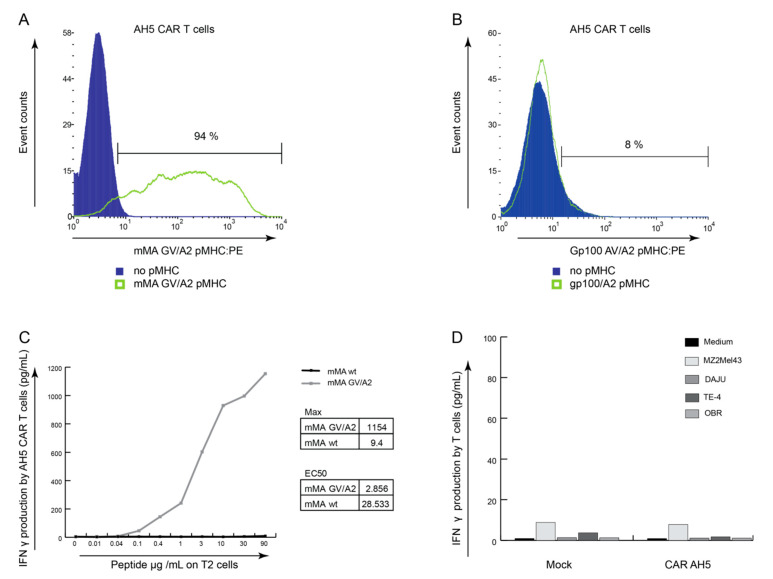
Anti mMA-GV/A2 antibody AH5, when expressed by T cells, recognises mMA-GV but not tumour cell-presented mMA-wt peptide. (**A**) AH5-CAR T cells bind specifically to mMA-GV/A2 complexes (green line). Filled (purple) histogram shows AH5-CAR T cells not stained with pMHC and were used as negative control. (**B**) AH5-CAR T cells do not bind irrelevant gp100/A2 complexes (green; filled (purple) histogram shows AH5-CAR T cells not stained with pMHC). (**C**) AH5-CAR T cells show increased IFNγ production when stimulated with T2 cells loaded with increasing concentrations of mMA-GV peptide. Non-loaded T2 cells yield no T cell IFNɣ production. EC50 for mMA-GV is 2.9 ug/mL or 2.2 mM and for mMA-wt 28.5 ug/mL or 26.1 mM; calculated by GraphPad Prism and both maximum and EC50 values are displayed. (**D**) AH5-CAR T cells show no to negligible IFNγ production when stimulated with tumour cells expressing mMA-wt peptide. Medium served as the negative control.

**Table 1 cancers-12-01255-t001:** Phage selections against wild type or heteroclitic mMA peptides complexed with HLA-A2.

Antigen^a^	Selection Round	Phage Input^b^	Phage Output^b^	Enrichment	(Pan-)MHC Binders^c^	TCR-like Binders^d^	Unique Patterns^e^
mM-A(GV)/A2 (combined w/ melanoma cells in round 2)	1	1 × 10^12^	9 × 10^6^	-	-	-	-
	2f	1× 10^12^	9 × 10^6^	1	-	-	-
	3	2 × 10^12^	5 × 10^8^	56	-	-	-
	4	6 × 10^12^	2 × 10^10^	2222	2/94 (2%)	80/94 (85%)	5
mM-A(GV)/A2 (combined w/ melanoma cells in round 4)	1	1 × 10^12^	9 × 10^6^	-	-	-	-
	2	1 × 10^13^	7 × 10^7^	7,7	-	-	-
	3	1 × 10^13^	2 × 10^10^	2222	57/282 (20%)	46/282 (16%)	18
	4f	2 × 10^13^	8 × 10^8^	89	6/282 (2%)	14/282 (5%)	9

^a^ See the MM section for a description of the different (combinatorial) libraries. Soluble biotinylated pMHC complexes were used to pull out binders. ^b^ Phage cfu’s determined by titration on E. Coli TG1 cells before and after each round of selection. ^c^ Phage-Fabs binding to at least two different pMHC complexes. ^d^ Phage-Fabs that bind only to pMHC complex to which they were selected. ^e^ Number of TCR-like antibody clones determined by BstNI restriction analysis. ^f^ Second or fourth selection round was performed using MAGE-A/HLA- A2-positive DAJU melanoma cells.

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
