# Peer review of "T Cells Expressing a TCR-Like Antibody Selected Against the Heteroclitic Variant of a Shared MAGE-A Epitope Do Not Recognise the Cognate Epitope"

_cancers, 2020, doi:10.3390/cancers12051255_

Round 1
Reviewer 1 Report
The study of Mesha Saeed and colleagues investigated whether MHC-restricted antibodies selected against heteroclitic or native peptides improve the therapeutic potential of gene-engineered T cells without loss of tumor specificity
The Authors study novel antibodies that recognize HLA-A2-restricted peptides that are shared among multiple melanoma associated antigens (MAGEs), thus providing a target for multiple tumors types. They have used a large human phage-Fab library to select TCR-like antibodies against the heteroclitic multi-MAGE peptide, and demonstrate that these antibodies show binding towards the modified pMHC complex but are not able to recognize native peptides nor antigen-positive tumor cells neither directly nor as a CAR that is expressed on the surface of T cells.
The Authors conclude that these findings do not warrant general use of heteroclitic peptides to establish immune therapies.
In this study it is not clear if the Authors used the same multi-MAGE (mMA) epitope described by Graff-Dubois et al. or a different shared epitope, because the results obtained by Graff-Dubois et al. are quite different. The Authors should discuss more deeply this issue.
There is still discordance among various studies, on the ability of heteroclitic peptides to induce T cell reactivity against wild type peptide presented by tumor cells.
Minor issues:
Line 99: 1nM and 50 nM should be corrected as 1 mM and 50 mM
Fig. 3B: histograms are not of good quality, especially for tumor cells only. Moreover the scale of event counts should be the same in the four panels.
Ref. 39 is missed. There are only 34 references listed in the manuscript
Author Response
Manuscript titled, “T cells expressing a TCR-like antibody selected against the heteroclitic variant of a shared MAGE-A epitope do not recognize the cognate epitope”, by Mesha Saeed, Erik Schooten Mandy van Brakel, David K. Cole, Timo ten Hagen and Reno Debets.
We want to thank the reviewer for their valuable time and their comments which we felt further improved the manuscript. Detailed point-by-point answers to the comments can be found below and are highlighted in the manuscript.
Response to Reviewer 1 Comments
Point 1: In this study it is not clear if the Authors used the same multi-MAGE (mMA) epitope described by Graff-Dubois et al. or a different shared epitope, because the results obtained by Graff-Dubois et al. are quite different. The Authors should discuss more deeply this issue.
Author’s reply to point 1:
Reviewer 1 correctly points to the unclarity regarding the identity of the mMA epitope used in our study versus the epitope used in the study by Graff-Dubois and colleagues. In fact, in the latter study various heteroclitic peptides were used to induce CD8 T cell clones, and the epitope that is identical to the mMA-GV used in our study is termed p248V9. Our manuscript already referred to this epitope (line 93), but additional referrals to this epitope are included in the revised manuscript in lines 95 and 381.
In an effort to explain why in contrast to our study, the study by Graff-Dubois shows that the mMA-GV-elicited T cell clones recognize the cognate epitope, we speculate along 2 major lines. First, we have generated an mMA-GV-specific CAR:28g that recognizes and mediates T-cell functions via an antibody-based receptor coupled to CD28 and Fc(e)RIg, which is inherently different than epitope recognition and intracellular signaling mediated via a natural T cell receptor (as was the case for the T cell clones generated by Graff-Dubois and colleagues). In fact, the AH5 CAR may display lower affinity for the native peptide and/or mediate suboptimal activation of T cell signaling components compared to a TCR. Secondly, our CAR T cells may display a less differentiated T cell effector phenotype compared to CD8 T cell clones, providing a potential other reason for compromised in vitro functions of the former T cells towards tumor target cells expressing the native mMA epitope.
From a more general standpoint, authors recognize that conflicting reports exist (Buhrman and Slansky 2013, Stuge et al 2004) regarding the usefulness of heteroclitic peptides in eliciting therapeutic T cell responses. To this end, we like to emphasize that our outcomes warrant caution and that induction of T cells against heteroclitic peptides provides no guarantee, whether it be for quantitative or qualitative reasons, that such T cells respond equally well against native peptides. For any T cell application, in particular therapeutic T cell applications, the T cell responses against native peptides should always be tested. The above interpretations of our results have been added to the revised discussion section from lines 386 and 452 onwards.
Minor issues:
Point 2: Line 99: 1nM and 50 nM should be corrected as 1 mM and 50 mM.
Author’s reply to point 2:
We thank Reviewer 1 for pointing this out, and we have corrected the manuscript accordingly.
Point 3: Fig. 3B: histograms are not of good quality, especially for tumor cells only. Moreover, the scale of event counts should be the same in the four panels.
Author’s reply to point 3:
We have improved the quality of histograms in Figure 3B, and have introduced the same scaling for all panels.
Point 4: Ref. 39 is missed. There are only 34 references listed in the manuscript.
Author’s reply to point4:
The missing reference, Buhrman and Slansky (2013), is now added to the revised manuscript.

Reviewer 2 Report
The manuscript described T cells expressing a TCR-like antibody selected against a heteroclitic variant of a shared MAGE-A epitope. Recently, CAR T-cell therapies, particularly adoptive cell therapies such as Kymriah and Yescarta, have risen to prominence. The authors revealed that prepared anti mMA-GV/A2 phage antibody AH5 recognized heteroclitic peptide but not native 305 mMA-wt/A2. Furthermore, AH5 CAR expressing T cells did not respond to tumor cells endogenously presenting mMA. Thus, the results will provide a novel development method based on CAR T-cell therapy to cure cancer. Therefore, the manuscript is not too excellent to be published. In other words, the manuscript is so excellent that it should be published.
Comments
(1) Can some of displayed chimeric antigen receptor become new antigens to usual immune system in living body?
(2) Are there any effective differences in introduced chimeric antigen receptors when they are recognized by tumor cells in vivo?
(3) Did introduced chimeric antigen receptors go through molecular evolution based on phage-Fab display library screeing? Can such optimized chimeric antigen receptors respond to arbitrary T cells?
(4) What kind of tumor cells did AH5 CAR expressing T cells respond to?
(5) In lines of 270 and 359, “didn’t” should be replaced with “did not”.
(6) In lines of 268-283, the heads and tails of sentences are accidented.
That is all.
Author Response
Manuscript titled, “T cells expressing a TCR-like antibody selected against the heteroclitic variant of a shared MAGE-A epitope do not recognize the cognate epitope”, by Mesha Saeed, Erik Schooten Mandy van Brakel, David K. Cole, Timo ten Hagen and Reno Debets.
We want to thank the reviewer for their valuable time and their comments which we felt further improved the manuscript. Detailed point-by-point answers to the comments can be found below and are highlighted in the manuscript.
Response to Reviewer 2 Comments
Point 1: Can some of displayed chimeric antigen receptor become new antigens to usual immune system in living body?
Author’s reply to point 1:
Authors thank Reviewer 2 for pointing out the immunogenicity of CARs. Indeed, due to the chimeric nature of CARs one cannot exclude the elicitation of an anti-CAR immune response. Already in an early clinical study with autologous T cells expressing a CAR directed toward carbonic anhydrase IX, an antigen highly expressed in renal cell carcinoma, we have reported distinct humoral and cellular anti-CAR responses in combination with limited peripheral persistence of transferred CAR T cells (Lamers C, JCO, 2006; Lamers C, Blood, 2011). Humoral responses were anti-idiotypic in nature and neutralized CAR-mediated T cell function, whereas cellular responses were directed against murine antibody variable domains. Of interest, such immune responses were not detected against boundary regions between antibody fragments and co-stimulatory and/or signaling domains from CD28 and/or Fc(e)RIg. Despite the potential occurrence and consequences of such immune responses, reported in multiple studies (Hege K, JITC, 2017; Stoiber S, Cell, 2019), it is important to note that with current designs, making use of completely human antibodies as is the case in our study, this risk is minimal. In fact, humanized libraries are increasingly being explored to lower the immunogenicity of CARs. Above text is added to the discussion section, line 413 and onwards.
Point2: Are there any effective differences in introduced chimeric antigen receptors when they are recognized by tumor cells in vivo?
Author’s reply to point 2:
CARs do show differences regarding their in vivo function depending on the CAR’s exact format, i.e., inclusion of linkers to space the antigen-recognition unit from the T cell’s plasma membrane, transmembrane domains and/or (multiple) intracellular co-stimulatory domains, such as CD28 (as is the case in the Yescarta product) or 4-1BB (as is the case in the Kymriah product). The varying formats of CARs as well as other aspects of cellular engineering, and their consequences for in vitro and in vivo T cell behavior are reviewed elsewhere (Gilham D, Trends Mol Med, 2012; Lim W, Cell, 2017); the take home message being that engineering of co-stimulatory CARs into the original TCR locus may provide additional safety and efficacy benefits. Part of above text is added to revised discussion section from line 483 onwards.
Point 3: Did introduced chimeric antigen receptors go through molecular evolution based on phage-Fab display library screening? Can such optimized chimeric antigen receptors respond to arbitrary T cells?
Author’s reply to point 3:
In the current study, the Fab AH5 did not go through molecular evolution. Since from earlier studies (Chames P, J Immunol, 2002) we have learned that such affinity enhancements may go at the expense of epitope-specificity, particularly in case CARs contain a co-stimulatory domain (Willemsen R, J Immunol, 2005), we have not performed such antibody evolutions.
Point 4: What kind of tumor cells did AH5 CAR expressing T cells respond to?
Author’s reply to point4:
AH5 was tested against a series of mMA-positive tumor cells, including melanoma, esophageal and bladder carcinoma cell lines, as listed in the revised Materials and Methods, line 118. These cell lines do express multiple MAGEs as well as HLA-A2 (Figure 3A), yet do not elicit a response by CAR T cells (Figure 5D). These results are presented and discussed in the revised manuscript.
Point 5: In lines of 270 and 359, “didn’t” should be replaced with “did not”.
Author’s reply to point 5:
We thank Reviewer 2 for pointing this out; this correction has been performed.
Point 6: In lines of 268-283, the heads and tails of sentences are accidented.
Author’s reply to point 6:
These lines are corrected in the revised manuscript.
